# A qualitative evaluation of a global surgery course within the University of Cape Town's master of public health curriculum: A cross-sectional study

Yvan Zolo[1]*, Wakisa Mulwafu[2,3], Moses Isiagi[1], Mary Kinney[1], Simon Le Roux[1], Salome Maswime[1]

1 Global Surgery Division, Faculty of Health Sciences, University of Cape Town, Cape Town, South Africa,
2 Kamuzu University of Health Sciences, Blantyre, Malawi, 3 The College of Surgeons of East, Central, and Southern Africa, Arusha, Tanzania

* zlsand001@myuct.ac.za, yvanzolovie@gmail.com

## Abstract

Access to safe surgery is a critical, yet often neglected, component of Public Health for health systems strengthening. Despite its importance, Global Surgery education remains absent from Public Health curricula, which may lead Public Health specialists unprepared to address the unmet global burden of surgical disease. This study evaluated a novel Fundamentals in Global Surgery course within the University of Cape Town's Master of Public Health (MPH) program, an initiative designed to bridge this educational gap. A qualitative descriptive study was conducted using in-depth structured interviews with ten program alumni (2022–2025). Data were analyzed thematically to explore the course's impact on participants' knowledge, careers, and professional identity. Analysis revealed five central themes defined by the following key insights: 1) a notable paradigm shift from viewing surgery as a clinical discipline to understanding it as a Public Health imperative; 2) the critical role of a multidisciplinary learning environment in mirroring real-world health systems; 3) the effectiveness of applied pedagogical approaches like case-based learning and intervention design; 4) a significant professional impact, including direct career pivots, advancements, and the cultivation of a practitioner-advocate mindset; and 5) logistical challenges of balancing coursework with professional duties. UCT's Fundamentals in Global Surgery course for MPH students demonstrates an effective model for integrating Global Surgery into Public Health education. By combining a systems-focused curriculum, an improvement science approach, and a multidisciplinary cohort, the course shows potentials to transform students into practice-ready change agents. The findings of this study inform a blueprint for curriculum development and strengthen the call to integrate Global Surgery into Public Health training.

**Data availability statement:** All relevant data are within the paper and its Supporting Information files.

**Funding:** The author(s) received no specific funding for this work.

**Competing interests:** The authors have declared that no competing interests exist.

## Introduction

Global Surgery is an indispensable field within global health, confronting inequities in surgical access that persist worldwide. For decades, surgical care remained absent from mainstream global health agendas, resulting in preventable suffering for millions. The World Health Organization (WHO) and The Lancet Commission on Global Surgery have highlighted the dire need for global efforts to improve surgical care, especially in low- and middle-income countries (LMICs) [1,2]. Addressing this gap requires not only clinical expertise but also Public Health approaches to strengthen surgical systems within broader health frameworks [3].

Since its formal establishment by the Lancet Commission on Global Surgery in 2015, the young but critical discipline of Global Surgery has seen academic programs proliferate. Global Surgery can be understood as the application of core public health principles—such as health equity, health systems strengthening, economics, and policy analysis—to the specific domain of the surgical ecosystem. It uses the surgical care continuum as a framework to teach and implement these universal public health concepts. Yet, its efforts are often fragmented; many well-intentioned interventions (e.g., short-term surgical missions and training camps) focus primarily on strengthening surgical skills in isolation [4–6]. This contrasts with the comprehensive approach of surgical systems strengthening, which targets the entire surgical care continuum—from prevention and pre-hospital care to anesthesia, surgery, and postoperative rehabilitation—to create resilient, equitable, and accessible systems [7]. This systems-based imperative underscores a critical gap in Public Health education, as the principles of surgical access and system strengthening remain absent from core curricula despite Public Health's fundamental focus on health systems, equity, and cost-effective policy. This omission leaves future Public Health leaders—the very professionals who will design and manage health systems—unequipped to address the massive surgical disease burden in LMICs, perpetuating the misconception that surgery is a costly luxury rather than an integral component of primary care and universal health coverage [8].

Traditionally, Public Health and Surgery have been separate. Hout et al. [9] identified significant knowledge gaps among Public Health students, including misconceptions that infectious diseases—rather than traumatic injuries—are the leading global cause of death, and that surgical care is economically unfeasible. These findings highlight a critical need for educational reform to better equip Public Health students with evidence-based Global Surgery knowledge, enabling them to effectively advocate for and implement surgical solutions in resource-constrained settings.

This study exploratorily assessed a Global Surgery course for MPH students by gathering alumni perspectives on its curriculum, teaching methods, and perceived professional impact.

## Materials and methods

### Ethics statement

Ethical approval was obtained from the University of Cape Town's Human Research Ethics Committee (HREC REF: 406/2025) prior to data collection. Verbal informed consent was

obtained from all participants prior to the interview. This method was approved by the University of Cape Town's Human Research Ethics Committee. Consent was documented by recording the affirmation of consent at the start of each interview session.

## Study setting and design

The University of Cape Town (UCT) has pioneered a dedicated Fundamentals in Global Surgery course within its Master of Public Health (MPH) degree, creating a formal academic pathway that integrates Global Surgery into Public Health education. This innovative curriculum is designed to provide students with comprehensive insight into how Global Surgery intersects with health policy, economics, and implementation science, equipping them to advocate for surgical system reform in resource-limited settings. We used a qualitative approach to explore the course's impact from the alumni perspective, examining gained competencies and their application in professional trajectories across low- and middle-income countries.

A qualitative methodology was selected as the most appropriate approach to address the study's exploratory aim. This design was chosen to facilitate a rich, in-depth understanding of the participants' personal experiences, perceptions, and the nuanced impact of the course on their professional journeys. Unlike quantitative surveys, this method allowed for the emergence of unanticipated themes and provided the depth of insight necessary to evaluate the complex, transformative outcomes of the educational intervention.

## Program and course description

The University of Cape Town's (UCT) Master of Public Health (MPH) is a multidisciplinary program based in the Faculty of Health Sciences' School of Public Health, a department committed to promoting equitable access to resources and highly competent healthcare to achieve a better quality of life for all. This professionally oriented degree is designed to be completed in approximately two years of full-time or four years of part-time study, requiring residence in Cape Town for its in-person coursework. The program's structure is built around a flexible yet rigorous curriculum where students complete 10 courses (or 8 courses for the Health Economics track) alongside a research-based mini-dissertation (or a more substantial minor-dissertation for Health Economics). Students deepen their expertise by choosing a specialization track, such as Epidemiology & Biostatistics, Health Systems, or the innovative Global Surgery track, which is newly available from 2025 and replaces the previous Community Eye Health track.

A pivotal component of the Global Surgery track is the "Fundamentals of Global Surgery" (CHM6045F) course. This compulsory first-semester module, which has also been available as an elective to all postgraduate students across the University of Cape Town since its inception in 2022, provides a comprehensive foundation for understanding surgical and anaesthesia care as a critical component of public health. The curriculum is meticulously structured to cover several core domains in a logical sequence. It begins by defining and quantifying the global and sub-Saharan African burden of surgical disease, establishing the scale of unmet need and its impact on populations. The course then systematically deconstructs surgical ecosystems, exploring the six World Health Organization building blocks of a health system as they apply specifically to surgical care. This includes deep dives into human resources for surgery, obstetrics, and Anaesthesia; infrastructure and essential equipment; financing models and the development of surgical benefit packages for universal health coverage; health information systems for surgical data tracking; service delivery at various levels of care; and the governance and leadership required for effective surgical systems planning. The curriculum extends into essential operational areas, delving into surgical research methodologies, quality improvement and patient safety frameworks, and the principles of advocacy, community engagement, and implementation science to ensure research evidence translates into effective, large-scale health interventions.

Taught through weekly, in-person two-hour sessions, the course employs a dynamic pedagogical approach. This includes real-world case-based learning that challenges students to solve problems endemic to resource-limited settings, guest lectures from leading global surgery practitioners like Professor Salome Maswime and Dr. Mary Kinney, and interactive discussions. The course is offered under the university's course-based fee structure, which allows students to

accurately calculate the cost of their studies. Upon completion, students emerge with the foundational knowledge, critical analysis skills, and practical tools needed to contribute meaningfully to global surgery through research, policy development, and the implementation of effective surgical healthcare programs.

### Participant selection and recruitment

A purposive sample of alumni (2022–2025) was recruited to ensure professional, geographic, and career-stage diversity. Potential participants were identified from the University's official records and subsequently contacted by email and phone. Sampling continued until thematic saturation was achieved, with a target of 8–12 participants representing 40–60% of the total alumni pool (N = 20).

### Data collection and management

In-depth interviews were conducted between the 6th August 2025 and the 2nd September 2025 using Zoom (Zoom Video Communications, Inc.), a secure, cloud-based videoconferencing platform [10]. The interview guide (provided as S1 Appendix) was developed by the main author and reviewed by Global Surgery education experts from the University of Cape Town's Global Surgery Division to ensure content validity and relevance to the study objectives. The interview guide underwent internal review and pilot testing to refine question clarity, eliminate ambiguity, and ensure thematic comprehensiveness. The finalized interview guide specifically probed participants' experiences with the *Fundamentals in Global Surgery course*, including the acquisition and application of global surgery competencies, and the course's impact on their professional practice and identity. Prior to each interview, verbal informed consent was obtained from all participants.

All interviews were conducted online by the main author in English and from a private setting to ensure confidentiality, and participants were likewise instructed to find a private location for the call to protect their privacy. Participants received instructions in advance for accessing Zoom. Sessions lasted 35–40 minutes each and were recorded with consent and encrypted. Verbatim transcripts were anonymized during transcription.

### Research team, reflexivity and data analysis

The research team included experts in Global Surgery, Public Health, and medical education. The interviewer was not involved in the design or instruction of the course curriculum.

Thematic analysis was conducted following Braun and Clarke's six-phase approach [11] using NVivo software [12]. This methodology enabled the identification of emergent themes regarding how the course was delivered and how alumni perceived, utilized, and were influenced by the competencies developed during the course. This study was conducted and reported in accordance with the Consolidated Criteria for Reporting Qualitative Research (COREQ) guidelines [13]. A completed COREQ checklist has been provided as S2 Appendix to ensure comprehensive reporting.

## Results

Ten alumni participated, representing 50% of graduates from the course. The cohort was diverse in age (20–49 years), gender (6 female, 4 male), and profession, including clinical (e.g., doctors, surgeons) and non-clinical (e.g., engineers, researchers) fields. Participants were from South Africa (n = 7), Malawi (n = 2), and the United Kingdom (n = 1), with experience ranging from students to senior professionals (16 + years) (Table 1).

Analysis of the ten interviews revealed five central themes regarding the alumni's experiences with the Fundamentals in Global Surgery course and the impact of the course on them.

### Theme 1: from clinical scarcity to health system improvement: a paradigm shift in perspective

The most profound and universal impact of the course was a fundamental reconceptualization of surgery's role in Public Health.

**Table 1. Participant Demographics and Professional Characteristics.**

| Participant Identifier | Age Group | Gender | Professional Category | Specific Professional Role | Country of Origin | Years of Professional Experience | Year of Course Completion |
|---|---|---|---|---|---|---|---|
| P1 | 30-39 | Male | Clinical practice | Medical Doctor | Malawi | 6-10 | 2024 |
| P2 | 20-29 | Female | Non-clinical practice | Biokinetics Specialist & Academic Researcher (Public Health) | South Africa | 0-5 | 2024 |
| P3 | 30-39 | Male | Non-clinical practice | Academic Researcher (Public Health and Global Surgery) | South Africa | 6-10 | 2022 |
| P4 | 20-29 | Female | Non-clinical practice | Academic Researcher (Public Health, Global Surgery and Policy) | South Africa | 0-5 | 2023 |
| P5 | 40-49 | Female | Non-clinical practice | Anthropologist, NGO leader | United Kingdom | 16+ | 2023 |
| P6 | 20-29 | Female | Clinical practice | Medical Student | South Africa | 0-5 | 2025 |
| P7 | 40-49 | Male | Non-clinical practice | Civil Engineer, NGO leader | South Africa | 11-15 | 2023 |
| P8 | 30-39 | Female | Clinical practice | Speech Therapist/Audiologist & NGO worker | South Africa | 6-10 | 2025 |
| P9 | 40-49 | Male | Clinical practice | Uro-gynecologist & Senior Academic Researcher (Global Surgery, Public Health and Policy) | Malawi | 16+ | 2022 |
| P10 | 20-29 | Female | Clinical practice | Medical Doctor | South Africa | 0-5 | 2025 |

**Sub-Theme 1A: dismantling the silo: surgery as a public health imperative.** Participants consistently described moving from a view of surgery as a purely clinical discipline to an integral component of Public Health and health systems. A medical student reflected: "*I used to see surgery as a siloed, last-resort intervention. Now I see it as integral to Public Health*" (P6). An Anthropologist and NGO director acknowledged surgery as a "*fundamental yet overlooked component*" of health systems (P7), while a speech therapist emphasized its role in "*equitable healthcare*" (P8). A senior researcher noted that the course "*cemented surgery as a Public Health priority*" requiring robust financing and governance frameworks (P9). A medical officer highlighted this reconceptualization: "*realizing that surgery is not only a clinical intervention but a fundamental component of Public Health*" (P10).

**Sub-Theme 1B: the tools for systems thinking: frameworks, economics, and sustainability.** Participants valued learning practical frameworks for systems thinking, policy, and implementation. One key insight was the economic burden of surgical disease: "*learning about the economic burden... made me realize how interconnected surgery is*" (P3). Others highlighted non-traditional concepts like green surgery, which expanded perspectives to include "*environmental implications*" and "*sustainability*" (P1). Innovative approaches such as citizen science were also noted (P8). A senior leader gained a "*structured framework to analyse and strengthen surgical systems,*" enabling him to "*translate research... into policy and practice*" (P9). A clinician found the "*surgical ecosystem*" framework essential for "*conceptualizing the vastness and complexity of Global Surgery*" and advancing "*beyond individual patient care*" (P10).

## Theme 2: the power of the multidisciplinary melting pot

The multidisciplinary environment was identified as a critical factor for understanding the complex, multi-stakeholder nature of surgical systems (Table 2).

**Sub-Theme 2A: learning *from* and *with* peers.** Participants cherished the opportunity to learn alongside clinicians, researchers, policymakers, and engineers. An engineer found working in "*multi-stakeholder teams with clinicians... directly*

**Table 2. Reported Professional Application of Course Learning.**

| Level of Impact | Type of Application | Description of Impact | Evidence of Application (Post-Course Activities) |
|---|---|---|---|
| **Strategic Leadership** | Systems Leadership & Scaling Innovation | Application of frameworks to design training programs, lead national initiatives, and implement data-driven service improvements at an institutional level. | P9: Designed a competency-based MPhil program; leads a national robotic training program; implemented clinical dashboards and ERAS protocols; uses GIS for equitable outreach planning.<br>P4: Coordinated the Global Surgery Summer School course at Stellenbosch University, applying frameworks from the UCT course to design content, and integrates learnings daily in ongoing Global Surgery research. |
| | Advocacy & Influence | Using knowledge to advocate for patients, systems change, or to educate peers, colleagues, and policymakers. | P5: "Asking questions and insisting on improvements" in NGO work. P1: Now advocates for Global Surgery. P9: Uses advocacy skills to engage ministries, hospital managers, and funders. |
| **Direct Career Impact** | Career Advancement/ Pivot | Course directly led to a new job, role, or defined career path. | P1: Transitioned to a Research Fellow in Global Surgery. P3: Applied for a Senior Chief Scientist role. P4: Shifted entirely to a Global Surgery research career. |
| | Research Leadership | Application of skills to lead or significantly contribute to new research projects. | P1: Leading projects on stillbirths and surgical care. P3: Leading a project on "failure to rescue". P4: Running independent projects on equitable access to healthcare and amputations. |
| **Integrative Practice** | Systems Thinking in Practice | Applying frameworks and principles to daily work to improve processes, design projects, and understand challenges. | P7: Better interprets clinicians' needs for infrastructure projects. P3: Uses frameworks for resource allocation in theatre. P5: Applies systems thinking to NGO leadership and business operations. P10: Started applying systems-thinking approaches in trauma care—for example, identifying process gaps in patient flow/transfer and discussing possible innovations with colleagues. |
| **Future Application** | Academic & Career Shaping | Using insights to shape academic pursuits and prepare for a future career path. | P8: Using insights to shape academic pursuits and prepare for a future career in health systems strengthening and advocacy. |

*applicable to my work*" (P7), while a researcher valued the "*practical exchanges*" (P3). This environment was particularly enlightening helping individuals overcome the "*imposter syndrome*" (P2) they initially felt. The presence of allied health professionals like P8 further enriched this diversity, and her suggestion to include even more perspectives from "*nurses and allied healthcare workers*" was a direct result of this experience.

**Sub-Theme 2B: learning from experts: the "Guest Lecturer" model.** The practice of having topics taught by field experts was highly praised. One participant noted, *"Policy was taught by someone with policy expertise, and HR management by someone experienced in that area"* (P4). This approach ensured that teaching was grounded in real-world experience and added credibility to the curriculum. This sentiment was strongly supported by P8, who praised the "*diverse guest lecturers providing a broad range of perspectives that enriched the learning experience*".

**Sub-Theme 2C: Building a professional network and community of practice.** Beyond the immediate learning, participants highly valued the establishment of a lasting professional network. This was described as a key benefit for navigating the niche field of Global Surgery. One participant stated they were "*grateful for the networks I've built and the continued support from colleagues*" (P3). For a senior leader, this was a specific goal, citing the desire to "*build connections with peers and mentors across Africa*" as a motivating factor for enrolling (P9).

## Theme 3: global surgery in action: pedagogical approaches to bridge theory and practice

The pedagogical approaches were valued for their effectiveness in translating abstract global surgery concepts into actionable skills.

**Sub-Theme 3A: case-based learning and interactive discussion.** Case-based learning was consistently highlighted as the most impactful pedagogical approach. Participants praised realistic scenarios, such as "*an accident in Khayelitsha,*"

for creating "*practical, interactive sessions*" that were "*the most engaging*" (P1). The small-group format successfully fostered "*critical thinking and practical problem-solving skills*" (P8), while also mirroring real-world interdisciplinary collaboration essential for "*surgical care and systems strengthening*" (P10).

**Sub-Theme 3B: applied assessments: from opinion pieces to intervention plans.** Applied assessments mimicking real-world tasks—particularly policy briefs and surgical intervention proposals—were highly valued. The final project was a "*culminating experience*" that synthesized learning into tangible outputs (P7). A senior participant emphasized the "applied orientation" was crucial, making it "*not just about learning, but about shaping tools*" for immediate use in leadership and research (P9). A clinician suggested enhancing this further with "*ready-to-use toolkits*" for designing system-level interventions (P10).

### Theme 4: professional impact: career confidence, catalysis, and creation

The course's impact extended far beyond the classroom, catalyzing significant professional growth (Table 2). The professional impact described by alumni was directly linked to the application of global surgery knowledge and frameworks in their work.

**Sub-Theme 4A: direct career impact.** The course directly catalyzed career advancement for many alumni. One clinician transitioned into Global Surgery research, joining a dedicated department (P1), while another gained the "*confidence and the tools*" to secure a Senior Chief Scientist role (P3). For a senior leader, it enhanced existing positions, consolidating leadership in urogynecology and increasing contributions to international guidelines (P9). A current student (P8) noted the impact differed by career stage.

**Sub-Theme 4B: the advocate's toolkit: skills for systems change.** Participants acquired a set of skills for driving change, moving from identifying problems to implementing solutions. *"Use advocacy skills to engage ministries... and funders." (P9); "Gave me the confidence... to contribute at a higher level." (P3); "Gave me... the language... to contextualize these... challenges."* (P10).

**Sub-Theme 4C: the advocate's mindset: confidence to challenge the Status Quo.** This new toolkit fostered a profound shift in mindset, empowering graduates with the confidence to question systemic flaws and insist on improvements. "*The course gave me a lot more confidence... and insist on improvements.*" (P5); "*I am... less willing to accept the first answer I'm given.*" (P5); "*The course... inspired me to consider how I can contribute to advocacy and policy changes...*" (P8); "*The course reignited my passion for global health.*" (P3); "*I have increasingly taken on informal leadership in quality improvement discussions...*" (P10).

### Theme 5: Logistical hurdles and curricular evolution

Despite the overwhelming positivity, participants identified challenges and offered constructive suggestions for improvement. Participants reflected on logistical challenges that, if addressed, could enhance the delivery and reach of the course.

**Sub-Theme 5A: the universal challenge of balance.** Balancing the course's demands with full-time work was the primary challenge, particularly due to lecture timing conflicting with clinical duties. This was less relevant for full-time students like P8, who reported "*no significant barriers.*" Senior professionals felt this acutely; one unit head cited "*heavy clinical and academic workload*" as a major barrier, managed through "*integrating coursework into live problems*" and strict "*time blocking*" (P9). A trauma clinician echoed this, identifying the "*biggest challenge*" as juggling clinical duties with academic work (P10).

**Sub-Theme 5B: suggestions for enhancement: depth, access, and connection.** Alumni proposed:
**Curricular Depth.** Expanding content on health economics, including the economic burden of surgical disease (P1), epidemiology (P2), digital health (P3), and citizen science (P8). P9 recommended adding modules on surgical financing,

digital health/AI, workforce planning, and global diplomacy. P10 advocated for more LMIC-specific case studies and content on surgical innovation and implementation science.

**Improved Pedagogy and Rigor.** Providing clearer guidelines on assignment expectations and raising the complexity of some tasks to better align with master's-level expectations (P8). P9 recommended more practical, ready-to-use resources: "*Add implementation labs with live datasets, ready-to-use toolkits (costing, business cases, stakeholder templates)*".

**Improved Access.** Providing readings and course outlines well in advance (P5) and formally exploring a hybrid delivery format to improve accessibility (P2, P6). Exploring a hybrid delivery format could also improve accessibility for working professionals (P2).

**Real-World Connection.** Organizing excursions or site visits to policy meetings or hospitals to strengthen the link between theory and practice (P4) and creating a formal mentorship program with alumni (P3) to strengthen the link between theory and practice.

**Broader Representation.** Incorporating perspectives from a broader range of healthcare professionals, such as nurses and allied healthcare workers (P8).

Participant impact was categorized into four levels (Table 2), illustrating how alumni have applied their learning professionally post-course. The table presents this framework, with the 'Evidence of Application' column providing specific examples serving as direct evidence for each level. Strategic leadership involved using frameworks to design programs and influence policymakers. Direct career impact included securing new roles in Global Surgery. Integrative practice meant daily use of systems thinking in operations. Future application captured how the course shaped academic and career goals for long-term systems strengthening.

## Discussion

### Summary of results

This qualitative evaluation of the University of Cape Town's pioneering Fundamentals in Global Surgery course unveils a unique educational approach that provides has pioneered Global Surgery education through a Public Health program, unlike most Global Surgery programs that are conventionally taught through Surgical disciplines. The findings demonstrate that its integration into Public Health education could pave the way for a new generation of practitioners armed with practical tools to act as change agents. Alumni narratives paint a picture of a transformative journey from conceptual understanding to possessing the skills needed to implement population-wide Global Surgery programs. To our knowledge, this was the first Global Surgery course integrated into an MPH program, and the UCT has recently launched an MPH Degree with specialization in Global Surgery, marking a unique and innovative approach to Public Health education.

### Synthesizing public health through a surgical framework

The course successfully used surgical systems as a practical framework for teaching the WHO health systems building blocks. The curriculum taught the principles of managing surgical services, focusing on core components: service delivery, financing, information systems, governance, human resources, and infrastructure. Participants described a paradigm shift from viewing surgery as a technical intervention to understanding it as a complex Public Health system, aligning with the foundational work of the Lancet Commission on Global Surgery. This was operationalized by allowing students to deconstruct a surgical ecosystem—analyzing its financing, workforce, and governance—to learn a transferable methodology for analyzing any surgical system [3,9].

### Global surgery as a public health specialization

This study reinforces Global Surgery as a vital MPH specialization, challenging perceptions of its incompatibility with Public Health [3,9]. Our findings highlight the value of integrating surgical systems into Public Health curricula, countering

views of surgery as purely clinical or cost-prohibitive [14]. by demonstrating that system strengthening, workforce planning, and innovative financing are cost-effective, equity-focused solutions aligned with Public Health's mission [15,16]. It helps dismantle the artificial divide between surgical care and Public Health practice [14,17,18] by illustrating how a Global Surgery curriculum can teach students to apply Public Health principles to the strengthening of surgical systems. Global Surgery is the application of Public Health principles—equity, efficiency, governance—to the surgical ecosystem [17].

The impact on alumni can be attributed to a curriculum designed to bridge theory and practice, [19–21] achieved through pedagogical shift from individual disease pathophysiology (clinical surgery) to population-level system pathology (Global Surgery). The course combined clinical realities with Public Health principles, providing an opportunity to transition from abstract theory to practice-ready capability, evidenced by alumni applying frameworks for resource allocation, program design, and policy change demonstrating the tangible outcomes of the program.

### Global surgery education

To situate this course within the broader ecosystem of public health education, it is helpful to distinguish it from other models. Unlike other Global Surgery courses designed for surgeons, and other clinicians [22,23], this model is situated within a public health curriculum. Compared to other MPH specializations—such as infectious diseases or health economics—which apply a public health lens to their respective domains, this course applies that same lens to the surgical system. Its distinct contribution is not in teaching new public health principles, but in applying them to a historically neglected yet critical component of universal health coverage. This positions it as a complementary, rather than competing, track within a comprehensive public health curriculum. This strategy directly addresses a critical gap identified by Francis et al. [24] who argue that the prevailing surgeon-dominated pipeline in Global Surgery must be expanded to include dedicated non-clinician scholars. This course provides a model for developing this essential workforce, equipping a new class of professionals to tackle the complex challenges of health economics, governance, and implementation science in surgical systems [25,26].

### Public health education

To contextualize this course within standard MPH frameworks, established MPH specializations are defined by their core analytical lens and health domain [26]. Within the common specialization of Global Health, which applies a broad, multidisciplinary lens to health challenges across national boundaries, sub-specializations naturally emerge to address specific priorities such as infectious diseases, nutrition, or maternal health. Established tracks in Public Health like Epidemiology apply a methodological lens to disease distribution, while Health Policy applies an analytical focus to governance and economics. A Global Surgery specialization is therefore defined by its application of these combined Public Health lenses—epidemiological, economic, and policy—to a specific and neglected domain: the surgical system [17,24,27]. It would thus produce a unique type of professional: a surgical systems architect, equipped to specifically diagnose, finance, and manage the surgical system as an integral component of universal health coverage. This formalization is needed to address awareness gap among Public Health professionals, who often lack comprehensive understanding of Global Surgery despite its important disease burden [25].

### Pedagogical innovations for systems-ready graduates

Essential pedagogical elements were key to the course's success. The multidisciplinary "melting pot" was a core component, forcing a collision of perspectives that broke down professional silos and mirrored real-world collaboration [20,21]. While multidisciplinary peer learning and instruction from field experts are established strengths of the broader UCT MPH curriculum, participants highlighted that these features were particularly critical for understanding the multi-stakeholder nature of surgical systems. The collaboration between clinicians and non-clinicians in this course provided a microcosm of the real-world collaboration needed to strengthen surgical care. Similarly, case-based learning and applied assessments

(e.g., designing intervention proposals) transformed students from passive recipients into active problem-solvers, synthesizing information to create feasible solutions.

The pedagogy of the course aligns with the principles of improvement science, as championed by the Institute for Healthcare Improvement and experts like Barker et al [28]. Its focus on actionable skills—advocacy, quality improvement, stakeholder engagement, and policy analysis—provides a practical, iterative framework for achieving measurable outcomes in complex systems. This moves beyond theoretical implementation science, equipping students with a tangible toolkit for change. By grounding this approach in Global Surgery challenges, the course serves as a blueprint for training practical, systems-ready improvement scientists.

### Cultivating the practitioner-advocate identity

A significant finding is the emergence of a new professional identity: the practitioner-advocate. This mindset represents the integration of Public Health education and systems thinking. Participants described a transition from powerlessness in the face of systemic dysfunction to an empowered mandate to challenge the status quo. Armed with evidence-based frameworks, they reported newfound confidence to engage stakeholders—embodying an embedded, courageous, and effective agent of change who leverages expertise to drive tangible progress toward health equity.

The reported professional impacts—such as career advancement and increased confidence—are core objectives of public health training. This evaluation suggests that the Global Surgery course successfully delivered on these objectives. However, we acknowledge that attributing these outcomes solely to this course is challenging, as they likely result from the cumulative effect of the entire MPH program.

### Limitations and methodological considerations

Our study has limitations. This study has several limitations. The sample, though representing 50% of program alumni, remains small. To our knowledge, the course was among the first of its kind integrated into a Master of Public Health program, which, while a strength in offering a novel case study, also means the findings are derived from a single, initial context. While data collection reached thematic saturation, self-selection and recall bias are possible, as participants self-reported impacts at a single timepoint, sometimes years after course completion. The findings primarily capture perceived competence among a predominantly clinical and research-oriented cohort; the perspectives of professionals from other key sectors like policy or finance are underrepresented. Furthermore, this study assesses educational and professional development outcomes but does not establish a direct link to tangible impacts on health systems or surgical care. Despite these constraints, the research provides valuable proof-of-concept and a transferable model for similar educational interventions.

### Recommendations for education, policy, and future research

Based on the findings of this study—though derived from a single institution—we propose the following recommendations. First, schools of Public Health explore integrating Global Surgery education into core curricula through dedicated specialization tracks. These tracks could teach standard health systems building blocks via the applied context of surgical care. Second, funders should invest in piloting these innovations to build non-clinical leadership for surgical systems and universal health coverage. Future research requires longitudinal, multi-institutional studies to track graduate impact, compare training models, and adapt the framework across diverse contexts.

### Conclusions

UCT's Fundamentals in Global Surgery course for MPH students demonstrates that the integration of surgical systems into Public Health education is more than a curricular addition—it is a valuable and effective educational strategy. The

course successfully uses the surgical ecosystem as an applied framework for teaching core public health principles, equipping a cohort of professionals with the knowledge and confidence to address surgical inequities. Rather than proposing a new paradigm, this model reinforces how established public health methods are essential for strengthening all components of a health system, including surgery. The course serves as a replicable blueprint for incorporating surgical system thinking into public health training, thereby helping to close a critical gap in the education of future health leaders. The success of the course also makes a case for a Master of Public Health with specialization in Global Surgery.

## Supporting information

**S1 Appendix. Interview Guide.**
(DOCX)

**S2 Appendix. COREQ Checklist: A 32-Item Checklist for Interviews and Focus Groups.**
(DOCX)

**S1 Transcripts. Anonymized Participant Interview Transcripts.**
(DOCX)

## Author contributions

**Conceptualization:** Yvan Zolo, Wakisa Mulwafu, Salome Maswime.

**Data curation:** Yvan Zolo, Wakisa Mulwafu, Moses Isiagi, Salome Maswime.

**Formal analysis:** Yvan Zolo, Wakisa Mulwafu.

**Investigation:** Yvan Zolo, Salome Maswime.

**Methodology:** Yvan Zolo, Wakisa Mulwafu, Salome Maswime.

**Project administration:** Yvan Zolo, Wakisa Mulwafu, Salome Maswime.

**Resources:** Yvan Zolo, Salome Maswime.

**Software:** Yvan Zolo, Moses Isiagi.

**Supervision:** Yvan Zolo, Wakisa Mulwafu, Salome Maswime.

**Validation:** Yvan Zolo, Wakisa Mulwafu, Moses Isiagi, Salome Maswime.

**Visualization:** Yvan Zolo.

**Writing – original draft:** Yvan Zolo, Wakisa Mulwafu, Moses Isiagi, Mary Kinney, Simon Le Roux, Salome Maswime.

**Writing – review & editing:** Yvan Zolo, Wakisa Mulwafu, Moses Isiagi, Mary Kinney, Simon Le Roux, Salome Maswime.

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
