## [Decision Letter · Decision Letter 0]

23 Oct 2025

PGPH-D-25-02795

EXPLORING THE IMPACT OF A NOVEL GLOBAL SURGERY PROGRAM INTEGRATED INTO THE MASTER OF PUBLIC HEALTH CURRICULUM: A QUALITATIVE CROSS-SECTIONAL STUDY

Dear Dr. ZOLO,

Thank you for submitting your manuscript to PLOS Global Public Health. After careful consideration, we feel that it has merit but does not fully meet PLOS Global Public Health’s publication criteria as it currently stands. Therefore, we invite you to submit a revised version of the manuscript that addresses the points raised during the review process.

Thank you for your submission.

Kindly address all reviewer comments, and apply appropriate changes, all of which are required for acceptance. One recurrent issue noted bu a couple of reviewers is overstatements in the text. Kindly review, and address as appropriate, and ensure that the conclusion is not an overreach, but supported by evidence within the study itself. For clarity, I am adding in a reviewer's comments from an attachment that also need to be appropriately addressed.

"The manuscript is clearly written, logically structured, and well-referenced. The topic of integrating global surgery into an MPH curriculum is timely and relevant, particularly within African public health training contexts. However, the paper significantly overstates the distinctiveness and real-world impact of this course. The described learning outcomes of peer learning, expert engagement, and professional development are not unique to this course. They align with the established design and objectives of the broader UCT MPH curriculum.

The claims of transformation and practical impact remain largely theoretical, with limited evidence of measurable outcomes or differentiation from other MPH specialisations. The course, while academically valuable, appears to deliver (in this early stage) a conceptual and knowledge-based enrichment rather than a transformative or practice-shifting experience. This is commendable.

However, I would suggest the authors provide a clearer comparative framing: how does this course differ from other UCT MPH courses / specialisations or from comparable global surgery or health systems programmes internationally? More importantly, how does its impact manifest relative to other specialisations i.e., what tangible professional or applied benefits would a student gain from selecting the Global Surgery track over others? Addressing these points would strengthen the paper’s credibility and help situate the course within the broader ecosystem of public health education, without overstating its influence.

**
General
**

The title reads as overstated and leans toward self-promotion. The term *“novel”* should be reconsidered or qualified. Including the institution’s name would also situate the work more appropriately and clarify context.Global surgery is somewhat of a misnomerIts hard to divorce away from the original nomenclature but its not accurate to speak of global surgery when in fact speaking about health systems / public health infrastructure / health economics… at best, surgery forms part… a critical and undermined part of the public health systemThe manuscript substantially overstates the real-world impact of the course. In the South African context, the practical contribution of a global surgery course to strengthening health systems or addressing inequities in access to care remains limited and largely theoretical. While the course may expand conceptual and academic knowledge, the claim that it meaningfully transforms health system outcomes or national policy readiness is not supported. If there is evidence, it would be great to include it to the framing. However, its unlikely that that could be attributed to this specific course.

A more balanced framing would acknowledge that this course complements, rather than substitutes for, the work of public health professionals engaged in implementation, program design, and system reform.

**
Specific
**

**Line 108: Sub-Theme 2A: learning *from* and *with* peers**

This finding is not unique to this course. Peer learning is a central pedagogical feature of the broader MPH programme itself. If anything, this sub-theme may reflect the strength of the MPH structure, rather than a distinctive feature of the global surgery course.

It would be helpful for you to clarify whether the course provides any particular type of peer or stakeholder interaction that differs meaningfully from the rest of the curriculum.

**Line 115: Sub-Theme 2B: Learning From Experts**

Similar to the comment above, this is a standard feature of most high-quality MPH programmes globally. Learning from experts is not a unique or innovative element of this course, and the manuscript should be careful not to present it as such.

**Line 144: Theme 4: Professional impact: career confidence, catalysis, and creation**

This theme of professional impact reads as overstated and its not sufficiently benchmarked. These outcomes are common goals of any MPH programme. Without comparative analysis i.e., against other MPH tracks at UCT or against comparable global surgery programmes such as Harvard’s Program in Global Surgery and Social Change, the conclusions posited here about professional impact lack a robust basis.

Confounding factors need to be acknowledged as well. Most of the sample size appears to have medical or academic backgrounds that predispose them to relevant professional advancement, independent of this specific course.

**Line 201: Table 2 Reported Professional Application of Course Learning**

The professional outcomes described could plausibly result from the broader MPH curriculum rather than this single course. Because the course is integrated within the degree structure, it is methodologically difficult to attribute impact solely to this course without an appropriate comparator.

A qualitative study of the stand-alone version of this course (which used to exist), prior to integration into the MPH, would provide a clearer and more defensible assessment of its distinct contribution.

**Line 240-243: “The foundational novelty if this course…diverse professional backgrounds.”**

The claim that diversity is a defining novelty is not supported by the sample of participants. Based on the data presented, none of the respondents appear to represent the broad non-medical or cross-sectoral backgrounds implied. The sample (predominantly clinicians, researchers, and allied health professionals) limits the ability to generalize or claim inclusivity as a defining strength. The manuscript needs to clarify whether course entry is selective and, if so, how diversity is operationalized.

As written, it is difficult to reconcile the intended goals of diversity and interdisciplinarity with the evidence provided.

**Line 292-298: “Based on the findings of this study…and adapt the framework across diverse contexts.”**

This conclusion is an overreach. Much of the global surgery framework draws heavily from established public health principles e.g., systems thinking, health equity, policy analysis, and resource allocation. What distinguishes it [global surgery] is the surgical framing of these established ideas, not a new paradigm. The manuscript risks positioning global surgery as both the hub and the spoke of public health discourse, when its contribution is better understood as contextual adaptation..."

In addition,address concerns on reviewers perception of the somewhat misleading title. Also present details of the course like the duration, curricular content, costs of the program, and so forth for the uninvolved reader.

Add in a qualitative study checklist e.g. SRQR etc and ensure to expand the informed consent section to clarify how that this was obtained from the interviewees. Please also provide the interview guide/questions as supplementary information. Was there any member checking?

We look forward to receiving your revised manuscript.

Kind regards,

Barnabas Tobi Alayande

Academic Editor

Journal Requirements:

Additional Editor Comments (if provided):

Reviewers' comments:

Reviewer's Responses to Questions

**Comments to the Author**

1. Does this manuscript meet PLOS Global Public Health’s publication criteria?

Reviewer #1: Partly

Reviewer #2: Yes

Reviewer #3: Yes

2. Has the statistical analysis been performed appropriately and rigorously?

Reviewer #1: N/A

Reviewer #2: N/A

Reviewer #3: N/A

3. Have the authors made all data underlying the findings in their manuscript fully available (please refer to the Data Availability Statement at the start of the manuscript PDF file)?

Reviewer #1: Yes

Reviewer #2: Yes

Reviewer #3: Yes

4. Is the manuscript presented in an intelligible fashion and written in standard English?

Reviewer #1: Yes

Reviewer #2: Yes

Reviewer #3: Yes

Reviewer #1: The manuscript is clearly written and ethically unproblematic, but it does not fully meet PLOS Global Public Health’s publication criteria in its current form. While it presents reflections on an academic course that may be of local educational interest, it does not constitute methodologically rigorous or generalizable research in public health.

The findings are descriptive rather than analytical, and the conclusions extend well beyond what the qualitative evidence can substantiate. As written, the study lacks sufficient methodological depth and comparative framing to meet the journal’s expectations for technical soundness or to justify its stated impact.

Reviewer #2: Thank you for the opportunity to review the manuscript "Exploring the impact of a novel global surgery program integrated into the master of public health curriculum". A few comments.

- The title is somewhat misleading, as the qualitative research has been conducted on the master's program and not only on the global surgery part (so it seems from the results). While the global surgery part should be emphasized, the title should be broader.

- The reader is not familiar with the program: a presentation of the duration, curricular content, costs of the program, and so forth should be described

- Was informed consent obtained from the interviewees?

- Please provide the interview guide / questions

- Except for theme 1, the described themes have no link to surgery so it seems- please indicate if the questions were specifically addressing global surgery or the program itself

Reviewer #3: Well done to the authors, this is a well written and well constructed qualitative study, addressing an area of growing importance: the integration of global surgery education within public health curricula. The topic is timely and highly relevant, given the increasing recognition of global surgery as a critical component of health systems strengthening and equitable healthcare delivery. The manuscript contributes meaningfully to the emerging literature.

The study is conceptually well-grounded, and the use of purposive sampling and thematic analysis is appropriate, however there could be a better paragraph on the justification of the methods chosen just for the reader. Participant characteristics are described clearly, providing good context for the interpretation of findings. The thematic analysis is well done, with clear, logical themes that capture both the perceived benefits and challenges of the course. The inclusion of long-term career and professional impact adds depth and value to the evaluation.

As some participants completed the course a few years prior to the interviews there is potential for recall bias. Participants' reflections may have evolved over time or been influenced by their subsequent professional experiences. It would strengthen the paper if the authors acknowledged this as an additional limitation and, where possible, discussed how it may have influenced findings.

**Do you want your identity to be public for this peer review?** For information about this choice, including consent withdrawal, please see our Privacy Policy

Reviewer #1: No

Reviewer #2: **Yes: ** Vicki Marie Butenschoen, MD, MSc

Reviewer #3: No

---

## [Decision Letter · Decision Letter 1]

30 Nov 2025

A QUALITATIVE EVALUATION OF A GLOBAL SURGERY COURSE WITHIN THE UNIVERSITY OF CAPE TOWN'S MASTER OF PUBLIC HEALTH CURRICULUM: A CROSS-SECTIONAL STUDY

PGPH-D-25-02795R1

Dear Dr. ZOLO,

We are pleased to inform you that your manuscript 'A QUALITATIVE EVALUATION OF A GLOBAL SURGERY COURSE WITHIN THE UNIVERSITY OF CAPE TOWN'S MASTER OF PUBLIC HEALTH CURRICULUM: A CROSS-SECTIONAL STUDY' has been provisionally accepted for publication in PLOS Global Public Health.

Best regards,

Barnabas Tobi Alayande

Academic Editor

Reviewer Comments (if any, and for reference):

Reviewer's Responses to Questions

**Comments to the Author**

Reviewer #1: All comments have been addressed

Reviewer #2: All comments have been addressed

publication criteria?

Reviewer #1: Yes

Reviewer #2: Yes

3. Has the statistical analysis been performed appropriately and rigorously?

Reviewer #1: N/A

Reviewer #2: N/A

4. Have the authors made all data underlying the findings in their manuscript fully available (please refer to the Data Availability Statement at the start of the manuscript PDF file)?

Reviewer #1: Yes

Reviewer #2: Yes

5. Is the manuscript presented in an intelligible fashion and written in standard English?

Reviewer #1: Yes

Reviewer #2: Yes

Reviewer #1: Thank you for your resubmission and addressing the comments.

Reviewer #2: All comments have been addressed

**Do you want your identity to be public for this peer review?** For information about this choice, including consent withdrawal, please see our Privacy Policy

Reviewer #1: No

Reviewer #2: **Yes: ** Vicki Marie Butenschoen
